# Next-Generation Probiotics for Inflammatory Bowel Disease

**DOI:** 10.3390/ijms23105466

**Published:** 2022-05-13

**Authors:** Marcella Pesce, Luisa Seguella, Alessandro Del Re, Jie Lu, Irene Palenca, Chiara Corpetti, Sara Rurgo, Walter Sanseverino, Giovanni Sarnelli, Giuseppe Esposito

**Affiliations:** 1Department of Clinical Medicine and Surgery, University of Naples “Federico II”, 80131 Naples, Italy; mapesc@hotmail.com (M.P.); sararurgo91@gmail.com (S.R.); sarnelli@unina.it (G.S.); 2Department of Physiology and Pharmacology “V. Erspamer”, Sapienza University of Rome, Piazzale Aldo Moro 5, 00185 Rome, Italy; alessandro.delre@uniroma1.it (A.D.R.); irene.palenca@uniroma1.it (I.P.); chiara.corpetti@uniroma1.it (C.C.); giuseppe.esposito@uniroma1.it (G.E.); 3Department of Anatomy and Cell Biology, China Medical University, Shenyang 110122, China; lvjie@cmu.edu.cn; 4Nextbiomics S.r.l., 80100 Naples, Italy; wsanseverino@sequentiabiotech.com

**Keywords:** probiotics, bioengineering, inflammatory bowel disease, biosensors, biotherapeutics

## Abstract

Engineered probiotics represent a cutting-edge therapy in intestinal inflammatory disease (IBD). Genetically modified bacteria have provided a new strategy to release therapeutically operative molecules in the intestine and have grown into promising new therapies for IBD. Current IBD treatments, such as corticosteroids and immunosuppressants, are associated with relevant side effects and a significant proportion of patients are dependent on these therapies, thus exposing them to the risk of relevant long-term side effects. Discovering new and effective therapeutic strategies is a worldwide goal in this research field and engineered probiotics could potentially provide a viable solution. This review aims at describing the proceeding of bacterial engineering and how genetically modified probiotics may represent a promising new biotechnological approach in IBD treatment.

## 1. Introduction

Probiotics are living microorganisms that may confer health benefit(s) to the host when supplied in adequate amounts [1]. Among their multiple effects, probiotics may help to maintain and/or restore a healthy community of microorganisms after being disturbed, release substances that regulate mucosal barrier integrity, and influence the host’s immune response mainly through the gut-associated lymphoid tissue (GALT). Several bacterial strains are currently used in real-life clinical settings to improve symptoms during inflammatory flare-ups and/or irritable bowel syndrome (IBS)-like symptoms occurring in IBD patients. Probiotics ameliorate the unbalance in microbiota populations, namely dysbiosis, which typically occurs in both patients and animal models of ulcerative colitis (UC) and Crohn’s disease (CD). Dysbiosis is strictly interrelated with several aspects of IBD pathophysiology, such as reduced mucin production, epithelial barrier integrity, opportunistic infections, and downregulation of anti-inflammatory bacterial products in the intestinal lumen [2]. Some probiotic strains, i.e., lactobacilli and bifidobacteria, can counteract pathogenic bacteria by producing antimicrobial peptides (bacteriocins and bacteriocins-like molecules). Furthermore, probiotic treatment can increase the concentration of short-chain fatty acids in the gut milieu, which exerts a protective role on the mucosa [3]. These and several other immunomodulatory effects of commensal bacteria prompted new clinical strategies to target intestinal microflora in IBD care (for a more comprehensive overview, readers are invited to refer to other reviews already published on the topic) [2,4]. In this framework, genetically engineered probiotics are one of the last frontiers for treating intestinal inflammatory conditions as they potentially display a dual effect: modulating intestinal dysbiosis, while also releasing therapeutically active molecules directly into the intestine, avoiding systemic drug administration and related side effects. The present review focuses on presenting an up-to-date overview of bioengineered probiotics in IBD treatment and highlights the main innovations in this field (Figure 1). Considering the current regulatory limitations to the genetically modified organism (GMO)-based therapies and the novelty of this methodology, the present review aims at underlining the most promising approaches currently under development for future clinical applications.

## 2. Microbiota Manipulation in IBD: Rationale and Current Limitations

IBD pathophysiology results from the interaction of four main players: genetics, environment, intestinal mucosa, and immunity. Interestingly, these factors have to face a shared variable, the intestinal microbiota that is, in turn, shaped by several environmental factors [5]. Indeed, the crosstalk between intestinal mucosa and microbiota is so imbricated that changes in mucosal homeostasis may trigger profound fluctuations in commensal bacteria populations and vice versa [6]. To date, distinct microbiota profiles have emerged in active IBD [7]. Both CD and UC patients display a decrease in *Firmicutes* and *Bacteroidetes* ratio and a rise in *Proteobacteria* and *Actinobacteria*. Specifically, *Bacteroides*, *Eubacterium*, *Faecalibacterium*, *Ruminococcus*, and other butyrate-producing genera are reduced in CD, enabling an imbalanced energy metabolism of the intestinal epithelium, leading to increased permeability [8]. Indeed, most of the depleted strains in IBD patients, such as *Bacteroides* and *Faecalibacterium* [9], can actively control intestinal homeostasis through the release of cytoprotective metabolites under physiological circumstances. These bacteria, in fact, ferment fibers to produce short-chain fatty acids, such as butyrate, that downregulate nuclear factor kappa-light-chain-enhancer of activated B cells (NF-κB)-related proinflammatory cascade and oxidative stress in intestinal epithelial cells and increase the anti-inflammatory mediators, such as interleukin-10 (IL-10) [10]. Moreover, butyrate restores the epithelial barrier integrity by promoting the direct proliferation of normal colonocytes, increasing the expression of tight junction proteins and antimicrobial peptides, and reducing intra-colonic pH [11]. Conversely, *Proteobacteria*, known to be increased in IBD, exert negative effects on the host physiology [12]. Carvalho et al. used an inflammation-prone mouse model, namely, the flagellin receptor Toll-like receptor 5 (TLR5)-deficient mice (T5KO), to study the role of microbiota in the development of intestinal inflammation. The authors found that mice developing colitis showed a definite microbiota signature characterized by increased levels of enterobacteria, especially of the Escherichia genus [13]. Even if the quality of the evidence is not always consistent and the subject is still a matter of debate, rising numbers of clinical and preclinical studies are highlighting the beneficial role of targeting the intestinal microbiota in IBD [14]. Administration of several probiotic strains proved to be directly capable of controlling and modulating the host immune response. For instance, *Escherichia coli Nissle 1917* (EcN) is a well-known immunomodulator capable of stimulating the increase in immunomodulatory cytokines, such as IL-10, and triggering an immunoglobulin A (IgA)-mediated immune response [15,16,17]. Moreover, *Bifidobacteria* and *Lactobacillus*, widely used as probiotics, exert a positive control on immune-mediated diseases (i.e., allergies) by limiting the release of IgE and acting on the Th1/Th2 ratio by leading to a switch to a Treg-mediated response [18]. Finally, other bacteria, such as *Lactobacillus Plantarum*, reduce proinflammatory cytokine production in mononuclear cells and prevent the adhesion of pathogens to the intestinal epithelial layer [19]. However, probiotic treatment in vivo mostly seems to act on symptom control, both in acute disease for symptom relief and in quiescent disease for the control of post-inflammatory IBS-like symptoms. Even if probiotics are a useful add-on therapy for IBD, real-world data demonstrate that probiotic treatment alone offers disappointing results in modulating intestinal inflammation in active IBD patients; hence, new technology focused on genetically upgrading bacteria could potentially revolutionize their use in IBD care [20].

## 3. Engineered Probiotics in IBD

With the outstanding advancements in genetic engineering, we recently witnessed the development of bacterial/probiotic strains genetically engineered to either act as “intestinal biosensors” (detecting inflammatory markers) or as “resident cells’ factories” of therapeutic molecules (biotherapeutics improving drug delivery at the mucosal surface) [21,22]. Most engineering approaches have focused on the transfection of plasmids encoding for immunoregulatory cytokines, reporter substrate, or anti-inflammatory mediators (Figure 1). Plasmids are small circular DNA molecules physically separate from chromosomal DNA and replicate independently. Their easy transfection into bacterial cells made them the major vector for inducing recombinant DNA expression into desired probiotic strains. Basic plasmid consists of essential DNA sequences that include a DNA replication origin, an antibiotic resistance gene (to select the transfected bacteria from non-transfected one), and the region referred to as the “Multiple Cloning Site”, in which exogenous DNA fragments are inserted by restriction enzymes. By recombinant DNA methods, the desired genes are spliced into a plasmid, which may also contain promoter regions to make plasmid expression chemically controlled (i.e., substrate and/or specific biomarkers). Both biosensors and biotherapeutic probiotics answer major unmet needs in IBD care. IBDs are chronic relapsing diseases in which a deep control of the mucosal inflammatory process (the so-called “mucosal healing”) represents the current goal of treatment. This, on one hand, implies that patients should undergo chronic immunosuppressive treatment, with undesirable systemic side effects, whilst, on the other, repeated invasive and costly investigations (i.e., endoscopies and biopsies) are often necessary to evaluate the presence of relapse and/or to detect residual signs of mucosal inflammation after treatment. Biosensors are live bacteria engineered to respond to specific biomarkers, indicative of inflammation, by producing a reporter substrate that could be easily detected (such as fluorescent proteins), hence limiting the need for invasive testing. To this aim, several biosensors have been recently developed to respond to markers of intestinal inflammation [22,23] or bleeding [21]. Yet, biosensors should display high sensitivity and specificity toward the selected biomarker to be used as diagnostic tools. This approach is, therefore, currently hampered by the limited knowledge of relevant biomarkers specific to gut inflammation and the number of characterized bacterial systems that can be reliably used as disease-responsive circuits. Given these shortcomings, most published studies have focused on developing engineered probiotics capable of expressing therapeutic molecules (biotherapeutic probiotics), either constitutively or “on-demand” through inducible systems that respond to exogenously administered substrates (commonly added to food or water) or to locally produced biomarkers (“sense and respond systems”) (Figure 2). All biotherapeutic probiotics are live bacteria designed to produce in situ anti-inflammatory molecules, offering the main advantage of achieving the topical release of therapeutics. Anti-inflammatory or immunosuppressant agents, indeed, may be released directly in situ, maximizing therapeutic concentrations in the target tissue using relatively smaller doses of the therapeutic compound, thus limiting systemic side effects. Constitutive systems offer the unquestionable advantage of using relatively simpler genetic modifications, mostly expressed in a constitutive fashion by the chosen probiotic platform. However, this comes to a cost considering the large amount of energy required for the probiotic to constitutively express these substances, while also exposing the risk of overproducing the therapeutic substance in unwanted sites, potentially hindering their effectiveness and safety, respectively.

Inducible probiotic systems overcome these two main concerns by producing the therapeutic substance in a controlled fashion only upon activation of inducible promoters. Depending on the considered probiotic construct, several types of inducible expression systems have been produced, capable of regulating the expression of the therapeutic molecule by adding exogenous substrates (such as xylan, palmitate, xylose, etc.) to animals’ food and/or water [24,25,26]. Sense and respond systems are an even more specific type of inducible system that combines and incorporates the technology of biosensors into live biotherapeutics to generate a more efficient and targeted delivery of the biotherapeutics only in response to a site-specific and/or disease-specific biomarker [21]. These systems do not respond to externally administered substances; rather, they “sense” specific environmental stimuli within the gut milieu (such as low pH, heat shock, or nitric oxide) and consequently release the therapeutic molecule, increasing the likelihood that its delivery is effectively site-specific and released under the most desirable circumstances (i.e., in the inflamed tissue) [27]. Nonetheless, as with biosensors themselves, sense and respond systems are hampered by the lack of specificity of biomarkers produced during gut inflammatory conditions and the limited number of reliable disease-responsive circuits identified in bacterial systems. In this context, a recent pivotal study, using a synthetic memory circuit in *Escherichia coli* (*E. Coli*), has allowed the recording of environmental stimuli differentially activated as the bacteria passed through the host and to retain this information via reporter gene expression, thus effectively enabling a noninvasive readout of transient signals generated under physiological and inflammatory circumstances, respectively [28]. The different genetic libraries obtained from healthy or dextran sulfate sodium (DSS)-treated mice, indeed, detected a number of activators or repressors differentially expressed during gut inflammation. Though their identity or function is not fully understood, this study marks an important step forward in our understanding of novel biomarkers that may be indirectly activated during the inflammatory process, paving the way to construct disease-responsive circuits by combining multiple redundant sensors responding differentially based on the localization within the gut. Most engineering approaches have focused on the transfection of plasmids encoding for immunoregulatory cytokines or direct anti-inflammatory mediators. In the following section, we will discuss the main findings in this field, highlighting the most used probiotic strains for bioengineering and comparing the different approaches.

### 3.1. Engineered Escherichia coli

*E. coli* strains, particularly EcN, are among the most widely used bacteria for genetic bioengineering. The extensive use of EcN is ascribable to the high genetic stability and non-transferability of the two small cryptic plasmids, pMUT1 and pMUT2 [29]. These two properties are essential in bioengineering protocols that employ plasmids to express exogenous genes in selected bacterial strains in order to achieve biological containment. EcN chromosome is also characterized by high genetic stability, since no changes in its sequence occurred, even after 100 sequential passages in vitro and through the intestinal tract of 14 newborn children treated with EcN for 24 months [30]. Adding to that, the use of EcN in bioengineering may lead to a double goal by merging its distinctive immunomodulatory effects on the mucosal immune system with the ability to produce and release bioactive anti-inflammatory molecules. Wang et al. demonstrated that engineered EcN releasing the immunoregulatory protein Sj16 derived from the helminth Schistosoma japonicum ameliorated DSS-induced colitis in mice by modulating the microbiota composition [24]. In this study, bacterial Sj16 peptide exerted protective effects against colitis by acting on peroxisome proliferator-activated receptor alpha (PPAR-α) receptors and restoring the population of *Ruminococcaceae* family, thus increasing butyrate levels in the intestinal lumen. A different approach entailed altering EcN metabolic pathway to induce the direct heterologous synthesis of (R)-3-hydroxybutyrate (3HB) [31]. Despite its function in energy supply, 3HB also has therapeutic potential by acting on macrophages reducing ILs production, and by restoring the Treg/Th17 balance. In this proof-of-concept study, the authors found that oral EcN synthesizing 3HB was superior to wild-type EcN or orally administered 3HB (100 mg/kg mouse weight) in improving mouse weights and colonic lengths, occult blood levels, and proinflammatory cytokine concentrations. An alternative explored approach consisted of the in-situ release of anti-inflammatory interleukins (ILs). Zhang et al. developed a constitutively expressing IL-35 E. coli as a novel oral delivery system displaying immunosuppressive effects, mediated by regulatory Tregs and B cells [32]. By downregulating Th17 cells, IL-35-producing E. coli proved to reduce the inflammatory response in a mice model of colitis. A different approach was undertaken by Cui and colleagues, who tested a light-responsive EcN secreting IL-10 in DSS-treated mice [22]. After oral administration, tissue-penetrable near-infrared (NIR) light was converted to blue light by up-conversion microgels to activate the recombinant EcN in the gut and to allow a controlled spatiotemporal release of IL-10 in mice. The authors found that IL-10-secreting EcN treatment suppressed intestinal inflammatory response and protected the intestinal mucosa against DSS-mediated injury. Furthermore, the concentrations of IL-10 produced by EcN were regulated under the NIR light and, thus, were expected to be safe. An intriguing and innovative approach was developed by Spisni et al. [33]. These authors used an engineered nonpathogenic invasive *E. coli* as an in vivo vehicle to transfect mammalian cells with anti-COX-2 RNA interference (RNAi). Owing to the expression of Invasin protein (Inv from *Yersinia pseudotuberculosis*), the engineered bacteria could invade intestinal epithelial cells without causing pathology and promote a functional transfer of RNAi from bacteria to the host. They designed an InvColi-based approach to efficiently silence the COX-2 gene and demonstrated that this system significantly reduced intestinal inflammation in UC and prevented the risk of associated colorectal cancer. This study effectively merged bioengineering techniques with gene therapy for the treatment of acute inflammation and long-term inflammatory-related carcinogenesis in UC.

### 3.2. Engineered Lactococcus lactis

Lactic acid bacteria (LAB) have been historically used in fermented foods and are generally regarded as safe. *Lactococcus lactis* (*L. lactis*) is a Gram-positive, spherical, homolactate, nonsporulating, and facultative anaerobic gut bacteria with hundreds of strains and variants published to date. Given the widespread use of *L. lactis* in food industries and its well-established safety in clinical and preclinical settings, this probiotic is a potentially useful tool in IBD treatment. *L. lactis* genome has been completely sequenced [34]; this aspect is crucial in the development of new bioengineering tools because of its easier manageability compared to other strains [35,36]. Besides its application in the food industry, *L. lactis* became one of the major LAB models in genetic engineering thanks to the development of genetic engineering tools, such as cloning and expression systems, that are compatible with its small-sized genome [37]. To date, *L. lactis* is widely used in IBD treatment and several bioengineered versions have been tested. Trefoil factors (TFF) and antitumor necrosis factor-α (TNF-α) nanobodies (single-domain antibody fragments) are among the therapeutic substances that have been constitutively expressed in *L. lactis* and tested in DSS-induced colitis in mice [38,39]. The TFFs are peptides that are differentially produced in specific sections of the gastrointestinal tract, displaying protective and reparative properties on the intestinal epithelium [38]. Specifically, TFF-3 is produced both in the small and large intestines, mainly by goblet cells. The in-situ production of TFF by *L. lactis* was shown to be more effective at healing colitis than the oral or rectal administration of the purified peptides. Another approach of *L. lactis* bioengineering was created to counteract the increased TNF-α release during colitis [39]. This *L. lactis* construct that releases the anti-TNF-α nanobodies displayed only the beneficial effects of anti-TNF therapy without adverse side effects. The smaller dimension of these nanobodies made them more stable compared to anti-TNF-α antibodies and less prone to provoking systemic immunological reactions owing to their topical use. A different study explored the use of *L. lactis* with the Microbial Anti-inflammatory Molecule (MAM)-encoding plasmid [40]. MAM is a peptide produced by *Faecalibacterium prausnitzii* that owes its anti-inflammatory properties to the downregulation of NF-κB in vitro. After the failed heterologous and chemical synthesis of MAM, the authors engineered *L. lactis* with a plasmid in which MAM’s cDNA was controlled by a eukaryotic promoter. The plasmid was transfected into intestinal epithelial cells and the MAM peptide was successfully produced in dinitrobenzene sulfonic acid (DNBS) and DSS mouse model of colitis. Like *E. coli*, *L. lactis* was also genetically engineered to express anti-inflammatory cytokines, namely IL-27 and IL-10. IL-27-producing *L. lactis* proved to be more effective than the administration of either IL-10-secreting *L. lactis* or IL-27 alone in resolving colitis in a preclinical mouse model [41]. It was shown that this strain increased the production of IL-10 in the intestinal epithelium, contributing to its effectiveness against colitis. Of note, *L. lactis* strains are the only engineered bacteria currently in phase II clinical trial for IBD (ClinicalTrials.gov Identifier: NCT00729872). In this setting, a human IL-10 (hIL-10)-producing *L. lactis* strain was tested for the treatment of moderately active UC and compared to hIL-10 given by injection and placebo. Unfortunately, even if hIL-10-producing L. lactis proved to be safe in humans, there was a non-superiority of the probiotic-treated group over the placebo [42].

### 3.3. Engineered Lactobacillus paracasei

To the best of our knowledge, only two studies by Esposito et al. provided evidence of the effectiveness of engineered *Lactobacillus paracasei* (LP) strains in models of colitis [25,43]. Using *Lactobacillus paracasei* subsp. *paracasei F19* engineered with human N-acylphosphatidylethanolamine-specific phospholipase D-(NAPE-PLD) gene (pNAPE-LP), the probiotic was able to selectively release palmitoylethanolamide (PEA) in the gastrointestinal tract under the boost of ultra-low doses of exogenous palmitate. PEA is an endogenous molecule belonging to the autacoid local injury amides (ALIAmides) family, which is locally produced on demand in response to cell and tissue damage [44,45]. PEA exerts potent anti-inflammatory effects, and it has been shown to improve intestinal inflammation, following both intraperitoneal and oral administration, in animal models of colitis [46]. More importantly, its efficacy has also been demonstrated in mucosal biopsies from patients with UC [47]. This strategy allowed for improved PEA delivery to the inflamed tissue and control of its release. Indeed, exogenous PEA is characterized by a disadvantageous oral pharmacokinetic profile. Its high lipophilicity limits its bioavailability, reducing its absorption. Thus, its oral administration (even in micronized and ultra-micronized forms) requires higher doses in humans or additives addition with potential side effects [48]. pNAPE-LP was able to release PEA directly on the mucosal surface. Moreover, the authors demonstrated that the expression of the NAPE-PLD enzyme in pNAPE-LP was directly related to the exogenous palmitate supply, generating a substate-dependent release bacterial platform.

### 3.4. Engineered Bifidobacterium longum

Bifidobacterium longum (*B. longum*) is a Gram-positive, catalase-negative, rod-shaped bacterium present in the human gastrointestinal tract [49]. Along with other *Bifidobacterium* species, it represents up to 90% of the bacteria in infant’s gastrointestinal microflora. Considering the high ratio of *Bifidobacteria* in infants’ microbiota, this probiotic could be potentially safe for developing an engineered probiotic platform, even in pediatric settings. One of the main advantages of employing *B. longum* strains as a probiotic is related to the high acid and bile tolerance, which enables its survival throughout the gastrointestinal tract and the successful colonization of the small and large intestines [50]. However, only a few studies tested the engineered *B. longum* in IBD. Van-De-Wetering et al. developed an engineered *B. longum* expressing the antioxidant enzyme manganese superoxide dismutase (MnSOD), which plays a key role in the pathophysiology of colitis [51]. MnSOD can improve colitis by reducing oxidative stress and reactive oxygen species (ROS) generation, inhibiting endothelial cell activation, and regulating adhesion molecule expression and leukocyte-endothelial interactions. The authors observed a significant decline in TNF-α, IL-1β, IL-6, and IL-8 levels following treatment, as well as an overall improvement of macroscopic and microscopic inflammatory markers. In another study, *B. longum* was genetically modified to express the α-melanocyte-stimulating hormone (α-MSH) (*B. longum*-pDGMSH2) in a DSS model of colitis [52]. α-MSH is a tridecapeptide derived from pro-opiomelanocortin that exhibits potent anti-inflammatory properties by downregulating the release of proinflammatory cytokines and mediators, such as ILs, TNF-α, and nitric oxide (NO), and upregulating anti-inflammatory cytokine, such as IL-10 [53]. *B. longum*-pDGMSH2 was able to restore epithelial barrier functions and protect by DSS-induced reduction in *Bifidobacteria* occurring during colitis. The main limitation concerning α-MSH use in IBD treatment is its extremely short life in vivo, and, to overcome the limitation, *B. longum* was successfully used here as a carrier to produce and deliver in situ α-MSH.

### 3.5. Engineered Bacteroides ovatus

As reported above, gut dysbiosis is a distinctive feature of IBD patients and *Bacteroides* represent some of the most reduced strains in both CD and UC [7,8]. Hence, using *Bacteroides ovatus* (*B. ovatus*) as an engineered probiotic platform could be an interesting and functional approach to restore *Bacteroides* populations and, simultaneously, effectively release bioactive compounds. However, to the best of our knowledge, a single study explored the use of engineered *B. ovatus* in the treatment of colitis [54]. Hamady et al. constructed a recombinant strain of *B. ovatus* for mucosal delivery of human transforming growth factor-beta (TGF-β) under the control of the dietary plant polysaccharide xylan. TGF-β is an epithelial cell growth factor involved in intestinal epithelium repair after mucosal injury [55]. *B. ovatus* is a xylanolytic colonic bacteria that release short-chain fatty acid through xylan breakdown [56]. The authors collocated the xylanase promoter upstream of the TGF-β coding sequence, generating a xylan-dependent system to release the active molecule. The oral treatment was effective in resolving the macroscopic pathological signs of colitis, as well as in reducing the expressions of pro-inflammatory markers. An overview of all biotherapeutic probiotics and their effects is provided in Table 1.

## 4. Bacterial Outer Membrane Vesicles: Potential New Tools in Probiotic Engineering

Outer membrane vesicles (OMVs) are nanostructures released by Gram-negative bacteria by membrane budding [59]. OMVs range in size from 20 to 300 nm and are released during all stages of bacterial growth. These vesicles are formed from the bacterial outer membrane (OM); thus, their composition closely resembles the components of the OM and periplasm, containing phospholipids, other bacterial outer membrane constituents, and periplasmic proteins trapped during membrane budding. These extracellular vesicles have been linked to many physiological processes, ranging from enhancing bacterial survival by detoxifying harmful compounds, enabling the formation of biofilms, and modulating host–pathogen interaction [60]. Natural OMVs indeed appear to be pivotal in bacteria–host cross-talk, ensuring the transmission of virulence factors into hosts’ cells, establishing colonization niches, and modulating the host immune response. For these characteristics, genetically engineering bacterial OMVs have been mainly explored in the development of novel engineered vaccine platforms. Nonetheless, bacterial OMVs are also promising new drug delivery vehicles, since they retain distinctive characteristics given their innate involvement in the intercellular exchange of biomolecules. Designing a probiotic able to load therapeutic compounds into extracellular OMVs, by using its endogenous packaging machinery, could allow exploitation of OMVs’ natural ability to target specific hosts’ cells and cross physical barriers (such as the epithelial intestinal barrier), while also increasing biocompatibility [61]. Naturally occurring OMVs from several types of commensal bacteria display protective effects on the host physiology that have been proven in murine models of IBD and could be exploited to improve drug delivery and immune regulation by engineered probiotics. *Akkermansia muciniphila* (AM) is an anaerobe Gram-negative whose relative abundance is decreased in IBD patients [62] and in experimental colitis [63]. Supplementation with AM-specific proteins from the outer membrane (Amuc_1100), secreted proteins (glucagon-like peptide-1-inducing protein), or extracellular vesicles could alleviate metabolic disease in humans and mice [64,65]. In parallel, unmodified OMVs secreted from both EcN and *Bacteroides fragilis* were shown to be able to ameliorate gastrointestinal inflammation in IBD or leaky gut models [66,67] (see Table 2) and have been engineered and used as antigen carriers [68].

To date, only a single study exploited engineered OMVs as a delivery platform for therapeutic factors in DSS-induced colitis [69]. This recent study by Carvalho et al. proposed a new engineering method to produce Keratinocyte Growth Factor-2 (KGF-2)-enriched OMVs of *Bacteroides thetaiotaomicron* (Bt). Bt is a member of the resident intestinal microbiota and one of the most significant butyrate-producing strains in the human intestine. The engineering method allowed Bt to incorporate bacteria-, virus-, and human-derived proteins into its OMVs. For the first time, this research presented a stable and effective drug delivery OMVs-based technology, paving the way to exciting future perspectives. In this study, the authors directly inoculated Bt- KGF-2-enriched OMVs in DSS-treated animals, observing a significant amelioration of DAI scores and macroscopical damage. In parallel, the microscopical damage score was reduced, while the number of goblet cells in the intestinal epithelium was restored. Regrettably, the authors did not provide a comparison between KGF-2 OMVs and KGF-2 alone. Despite this limitation, Carvalho et al. provided an innovative method to efficiently control inflammation by using bioengineered OMVs in murine colitis.

## 5. Conclusions and Challenges

Overall, a growing number of research groups are focusing on designing engineered probiotics for the treatment of inflammatory bowel disorders. New tools in genetic engineering have allowed the design of sophisticated systems that could sense intestinal inflammatory markers and/or produce topically therapeutic molecules. There is undoubtedly a strong rationale underlying the role of microbiota in IBDs and mounting evidence pinpoints the intestinal microflora as a key player in modulating the host immune response. However, despite the promising results in preclinical models, one must consider that the mice microbiota is radically different from the human one [70], and several encouraging preclinical approaches have failed when translated to humans. There is unfortunately a lack of evidence in clinical models, owing to the novelty of the methodology and the safety concerns regarding GMO-based therapy. To date, a single clinical study has tested an engineered probiotic in phase II clinical trials in humans and failed to prove a benefit over the placebo. This could be explained by several factors that could potentially hinder the effectiveness of engineered probiotics when introduced into the host microbiome. Indeed, analogously to native probiotic treatments, genetically engineered probiotics could fail to achieve colonization due to the complexity of establishing a niche to survive in the gut microbiome. Moreover, the difference between human and mouse microbiota may lead to increased genetic mutations or a reduction in the growth rate of the genetically engineered bacteria due to the different environments. Furthermore, the increased energetic burden caused by synthetic circuits in engineered bacteria could hamper their bioavailability and efficacy (particularly for constitutive systems). Another crucial point regards containment, specificity, and safety issues of these genetically engineered microorganisms. Once transferred to the host, these microorganisms have been incapable of transferring their modified DNA to the environment, with unpredictable outcomes. At the same time, engineered probiotics should release bioactive molecules only when/where needed, confining their activity to the target tissue(s). To this end, sense and respond systems, capable of releasing the biotherapeutics only in response to biomarkers of inflammation, hold great prospects in maximizing their efficacy, while answering safety concerns. Although some progress has been made in this context, we are still far from fully appreciating the complexity of bacteria–host interactions and consequentially exploiting it to deliver biotherapeutics (Figure 3). Even so, current advances in developing live biotherapeutics indicate that they are promising treatments for IBD. With synthetic biology tools, scientists can rapidly create various genetic circuits and, by high throughput screening, a multitude of options can be evaluated. Among prebiotics, phytochemicals, such as dietary polyphenols (phenolic acids, flavonoids, stilbenes, tannins, and diferuloylmethane), are promising candidates for IBD therapy in pre-probiotic formulations, since these are already proven effective in the treatment of several chronic diseases, such as type 2 diabetes [71,72]. Engineered probiotics that convert multiple unabsorbed dietary compounds into molecules that influence intestinal microbiota and host immunity might reveal important tools in the treatment of IBD. Next-generation probiotics could represent the future of personalized medicine in IBD care, allowing patients to be treated in a more tailored manner, through disease-responsive circuits. If this technology fulfils its promises, we could be at the dawn of a new era in personalized IBD therapy.

## Figures and Tables

**Figure 1 ijms-23-05466-f001:**
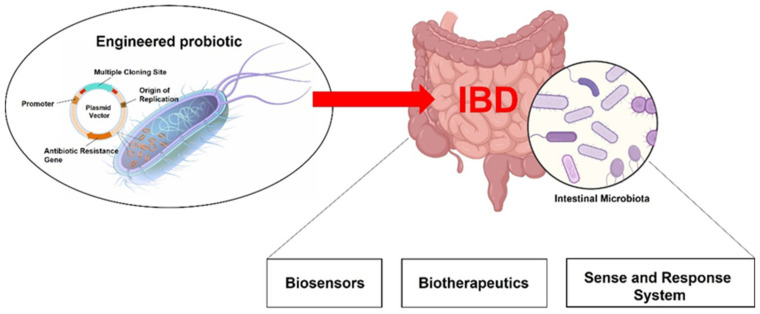
Live biotherapeutics as new treatments for inflammatory intestinal diseases. Advances in the use of probiotics led to the development of engineered bacteria that act as intestinal biosensors, which detect specific biomarkers and work as diagnostic tools, and/or drug delivery systems able to release therapeutic substances directly into the intestinal lumen. Plasmids are the major vector for inducing recombinant DNA expression into desired probiotic strains. Despite diverse plasmids encoding for immunoregulatory cytokines, reporter substrate, or anti-inflammatory mediators being engaged in probiotics bioengineering, distinct regions are generally recognized: DNA replication origin, an antibiotic resistance gene, and the “Multiple Cloning Site” where exogenous DNA fragments are inserted by restriction enzymes.

**Figure 2 ijms-23-05466-f002:**
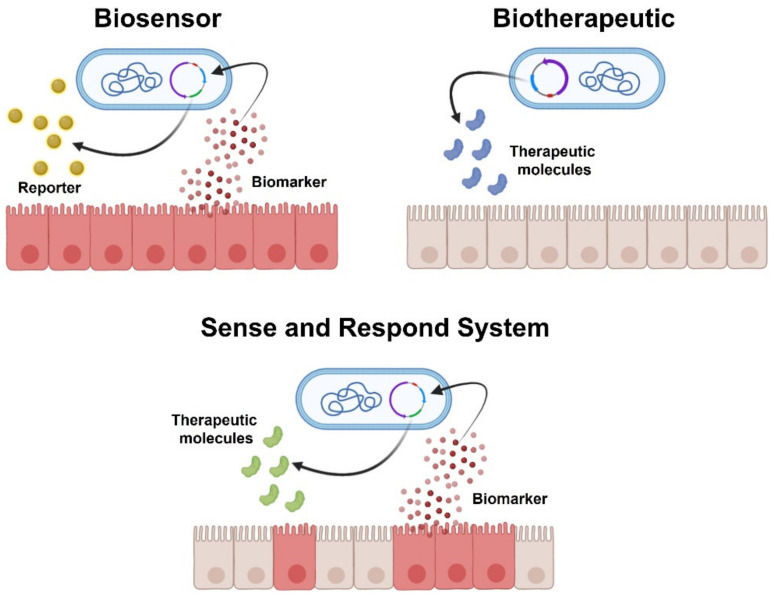
Main approaches in probiotics engineering exploited in IBD. Biosensors can induce the expression of a reporter (usually a fluorescent marker) upon detecting specific biomarkers of inflammation. Biotherapeutics are able to produce at the mucosal surface a therapeutic molecule either constitutively or following the activation of an exogenous substrate (inducible systems). Sense and respond systems incorporate the technology of biosensors by responding to specific biomarkers of inflammation with the production of a therapeutic molecule.

**Figure 3 ijms-23-05466-f003:**
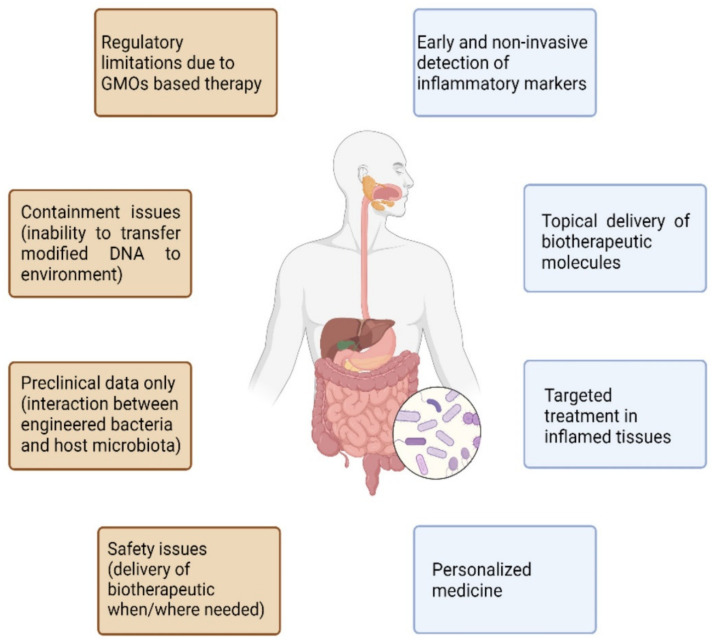
Current advances and concerns in developing live biotherapeutics.

**Table 1 ijms-23-05466-t001:** Summarizing table of the main engineered strains for IBD treatment.

Strain	Therapeutic Factor	Effect	Target/Model	Reference
*Escherichia coli Nissle*	(R)-3-hydroxybutyrate	SCFA levels **↑** *Akkermansia* spp. **↑** Microscopical/macroscopical damage and DAI scores **↓** ILs levels **↓**	Colitis/DSS induced Colitis	[31]
	*Schistosome* immunoregulatory protein	*Ruminococcaceae***↑** butyrate and retinoic acid production **↑** Treg cells **↑** Th17 cells **↓** Microscopical/macroscopical damage score **↓**	Colitis/DSS induced Colitis	[24]
	IL-35	CD4^+^ IL-17A^+^ Th17 cells **↓** CD4 ^+^ CD25^+^Foxp3^+^ Tregs cells in spleen and mesenteric lymph nodes **↑** Colonic and serum IL-10 and IL-35 **↑** Colonic and serum IL-6 **↓** Microscopical/macroscopical damage score **↓** DAI score **↓**	Colitis/DSS induced Colitis	[32]
	IL-10	Microscopical/macroscopical damage score **↓** IL-1, IL-12, MPO, and TNF-α levels **↓**	Colitis/DSS induced Colitis	[22]
	*Inv*Coli anti-COX-2 RNAi	DAI score **↓** Microscopical/macroscopical damage score **↓** ILs levels **↓** colitis-associated shift of gut microbiota **↓** COX-2 expression **↓**	Colitis/DSS induced Colitis	[33]
*Lactococcus lactis*	IL-27	DAI score **↓** Microscopical/macroscopical damage score **↓** ILs levels **↓** CD4(+) and IL-17(+) T cells in gut-associated lymphoid tissue **↓**	Colitis/transfer of CD4(+) CD45RB(hi) T cells into Rag(−/−) mice	[41]
	TNFα-neutralizing Nanobodies	Microscopical/macroscopical damage score **↓** MPO levels **↓** Do not interfere with systemic *Salmonella* infection in colitic IL10^−/−^ mice.	Colitis/DSS induced Colitis and IL10^−/−^ mice	[39]
	Trefoil factors	Ptgs2 expression Microscopical/macroscopical damage score **↓** MPO levels **↓**	Colitis/DSS induced Colitis and IL10^−/−^ mice	[38]
	IL-10	Microscopical/macroscopical damage score **↓**	Colitis/DSS induced Colitis	[57]
		Microscopical/macroscopical damage score **↓** IFN-γ levels **↓** IFN-γ, IL-12, IL-17 positive cells ↓	Crohn’s Disease/TNBS induced gut inflammation	[58]
	pILMAM (cDNA coding for *Faecalibacterium-prausnitzii*-derived Microbial Anti-inflammatory Molecule)	IL-17, IL-6, and IL-5 levels ↓ IL10, TGFβ levels **↑** DAI score ↓	Colitis/DSS and DNBS induced Colitis	[40]
*Bifidobacterium longum*	MnSOD	Microscopical/macroscopical damage score ↓ MPO levels ↓ TNF-α, IL-1β, IL-6 and IL-8 levels ↓	Colitis/DSS induced Colitis	[51]
	α-MSH	Microscopical damage score ↓ NF-kB activation, TNFα, NO, and IL-6 levels ↓ MPO levels ↓ IL10 levels ↑	Colitis/DSS induced Colitis	[52]
*Lactobacillus paracasei* subsp. *Paracasei F19*	NAPE-PLD/Palmitoylethanolamide	DAI score ↓ Microscopical/macroscopical damage score ↓ MPO levels ↓ iNOS and COX-2 expression, IL-1β, PGE-2, TNFα, and NO levels ↓ tight-junction protein expression in the gut epithelium ↑	Colitis/DSS induced Colitis	[25]
		Microscopical/macroscopical damage score and MAC387^+^ cells ↓ MPO levels ↓ TLR4, RhoA GTP, p-p38 MAPK, NF-kB, HIF1-α expression ↓ tight-junction proteins expression in the gut epithelium ↑	Colitis/TCdA induced Colitis	[43]
*Bacteroides ovatus*	TGF-β	DAI score ↓ Microscopical/macroscopical damage score ↓ IL-1β and TNFα levels, and COX-2 expression ↓	Colitis/DSS induced Colitis	[54]

SCFA: short-chain fatty acid; DAI: disease activity index; DSS: dextran sulfate sodium; ILs: interleukins; MPO: myeloperoxidase; COX-2: cyclooxygenase-2; Ptgs2: prostaglandin-endoperoxide synthase 2 gene; IFN-γ: interferon-γ; pILMAM: interleukin and microbial anti-inflammatory molecule (MAM)-encoding plasmid; TGFβ: transforming growth factor β; TNBS: 2,4,6-trinitrobenzene sulfonic acid; DNBS: dinitrobenzene sulfonic acid; TNF-α: tumor necrosis factor alpha; MnSOD: manganese superoxide dismutase; NF-kB: nuclear factor kappa-light-chain-enhancer of activated B cells; NO: nitric oxide; iNOS: inducible nitric oxide synthase; α-MSH: α-melanocyte-stimulating hormone; PGE-2: prostaglandin E2; NAPE-PLD: N-acylphosphatidylethanolamine-specific phospholipase D; TLR-4: Toll-like receptor 4; RhoA: Ras homolog family member A; HIF1- α: hypoxia-inducible factor 1-alpha; p-p38 MAPK: phospho-p38 mitogen-activated protein kinases. ↓: decrease; ↑ increase.

**Table 2 ijms-23-05466-t002:** Table summarizing the main OMV applications in IBD.

Strain	Therapeutic Factor	Effect	Target/Model	Reference
*Akkermansia muciniphila*	OMVs	DAI score ↓ Microscopical/macroscopical damage score ↓	Colitis/DSS induced Colitis	[63]
	OMVs	Microscopical/macroscopical damage score ↓ epithelial permeability in vitro and in vivo ↓ tight-junction proteins expression in the epithelium in vitro and in vivo ↑	low-grade inflammation and leaky gut/HFD-induced leaky gut in vitro leaky gut model	[65]
	OMVs	tight-junction proteins expression ↑ TLR4 and TLR2 expression ↓	IBDs and metabolic syndrome/Caco-2 culture	[64]
*Bacteroides fragilis*	OMVs	Microscopical/macroscopical damage score ↓ Foxp3 and IL-10 from CD4+ T cells ↑	Colitis/DSS induced Colitis	[67]
*Escherichia coli Nissle 1917*	OMVs	DAI score ↓ Microscopical/macroscopical damage score ↓ ILs, iNOS, and TNFα levels ↓ IL10 levels ↑ tight-junction proteins expression in the gut epithelium ↑	Colitis/DSS induced Colitis	[66]
*Bacteroides thetaiotaomicron*	engineered OMVs containing KGF-2	DAI score ↓ Microscopical/macroscopical damage score ↓ Protection and restoring of globet cells↑	Colitis/DSS induced Colitis	[69]

OMVs: outer membrane vesicles; DAI: disease activity index; DSS: dextran sulfate sodium; TLRs: Toll-like receptors; TNF-α: tumor necrosis factor-alpha; iNOS: inducible nitric oxide synthase; KGF-2: Keratinocyte Growth Factor-2. ↓: decrease; ↑ increase.

## Data Availability

Not applicable.

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
