# Peer review of "Next-Generation Probiotics for Inflammatory Bowel Disease"

_ijms, 2022, doi:10.3390/ijms23105466_

Round 1

Reviewer 1 Report

In their review titled “Next-generation probiotics for inflammatory bowel disease” Pesce et al have made a commendable effort in their review discussing the new possibilities of probiotics for IBD management or treatment. They discuss the various options for the use of probiotics as biosensors, biotherapeutics and sense and respond systems. They have also given a tabular summarized version of the strains which makes it easier on the reader.

Minor comment

  1. It is suggested to start off the review with the definition of probiotics-Probiotics are live microorganisms that are intended to have health benefits when consumed or applied to the body (NIH-https://www.nccih.nih.gov/health/probiotics-what-you-need-to-know).

Author Response

In their review titled “Next-generation probiotics for inflammatory bowel disease” Pesce et al have made a commendable effort in their review discussing the new possibilities of probiotics for IBD management or treatment. They discuss the various options for the use of probiotics as biosensors, biotherapeutics and sense and respond systems. They have also given a tabular summarized version of the strains which makes it easier on the reader.

Thank you for your very kind feedback. We are pleased that the reviewer felt that the initial draft contained important information and we appreciate the reviewer’s comment on how to improve our review presentation. We have made the suggested revision as indicated below.

Minor comment

  1. It is suggested to start off the review with the definition of probiotics-Probiotics are live microorganisms that are intended to have health benefits when consumed or applied to the body (NIH-https://www.nccih.nih.gov/health/probiotics-what-you-need-to-know).

We agree with the Reviewer that this sentence should be revised for general definition, and we revised the sentences from P1 line 28 accordingly.

Reviewer 2 Report

The authors tried to make a Review on the Next-generation probiotics for inflammatory bowel disease.

At a first glance, 11 pages (in the MDPI format) cannot for sure to describe this vast topic that the authors have chosen. Aspect of the paper needs also extensive improvement. In the actual shape, I find no relevance of the content. There are many papers on the same topic, already published, much more extensive and better done, more detailed, etc. Please see below my main suggestions regarding the improvement of his manuscript:

General concerns

  1. Remove the empty spaces between the paragraphs, in the same section. The text in each section must be compact (please see the Instructions for authors in this regard), and the aspect of the manuscript more professional.
  2. All Latin names must be written in italics, respecting the international rules.
  3. Acronyms/Abbreviations/Initialisms have been defined the first time they appear in each of three sections: the abstract; the main text; under the first figure or table. When defined for the first time, the acronym/abbreviation/initialism should be added in parentheses after the written-out form. Please check and revise the entire manuscript in this regard.
  4. For a Review type paper, long paragraphs are not supported at all by references. Please complete. (i.e. L167-178, L213-218, and so on). If there are “most studies”, they must be exemplified.

Specific concerns

  1. Under Table 1, please detail all Abbrev. Used in the table, according to the Instructions for authors. Same for Table 2.
  2. Please better describe the process of engineered bacteria and the modalities of this process; how it can be technically done?
  3. Please detail the role of certain natural compounds that can be delivered together with the probiotic and decrease the pro-inflammatory markers, such as certain plant-based compounds or supplements that act as prebiotics - in this regard, I suggest checking and referring to https://doi.org/10.3390/diagnostics11061090 https://doi.org/10.3390/microorganisms9030618
  4. Some figures must be done, to summarize main sections, to increase the impact of your article, and its aspect as well. Detail the mechanisms also.
  5. A special figure should be made regarding the potential side effect of these modified bacteria in IBD or for the general health.

Author Response

The authors tried to make a Review on the Next-generation probiotics for inflammatory bowel disease.

At a first glance, 11 pages (in the MDPI format) cannot for sure to describe this vast topic that the authors have chosen. Aspect of the paper needs also extensive improvement. In the actual shape, I find no relevance of the content. There are many papers on the same topic, already published, much more extensive and better done, more detailed, etc. Please see below my main suggestions regarding the improvement of his manuscript:

We agree that improving the presentation with additional figures would be very beneficial and we added two figures to summarize the main points covered by our review (new Figure 1) and the advantages/side effects related to engineered probiotic use (Figure 3), respectively. We also agree that improving the precision of certain words/acronyms/abbreviations would increase the overall clarity of the review and we have revised much of the text in this regard. We can definitely see the reviewer’s point and understand the feeling that describing the process of engineered bacteria would be beneficial for readers. We added text to the “Engineered probiotics in IBD” section and Figure 1 to illustrate the main engineering approach that is commonly used in this area. Also, we have added additional references (ref. #71 and 72) to highlight recent works by others that summarize the newest evidence supporting the probiotic and prebiotic use in IBD therapy. Thank you so much for your very insightful comments. These have been an immense help in our efforts to improve this manuscript. The reviewer has many important points, and we feel that addressing these resulted in major improvements to the content and presentation of the review. How we addressed each point is detailed in our response below and noted by highlights in the revised text.

General concerns

  1. Remove the empty spaces between the paragraphs, in the same section. The text in each section must be compact (please see the Instructions for authors in this regard), and the aspect of the manuscript more professional.

Thank the reviewer for this suggestion. We revised the manuscript accordingly.

  1. All Latin names must be written in italics, respecting the international rules.

We completely agree and have modified the manuscript accordingly.

  1. Acronyms/Abbreviations/Initialisms have been defined the first time they appear in each of three sections: the abstract; the main text; under the first figure or table. When defined for the first time, the acronym/abbreviation/initialism should be added in parentheses after the written-out form. Please check and revise the entire manuscript in this regard.

We completely agree and have revised the manuscript accordingly

  1. For a Review type paper, long paragraphs are not supported at all by references. Please complete. (i.e. L167-178, L213-218, and so on). If there are “most studies”, they must be exemplified.

From line 167 to line 173, we discussed the extensive results related to Naydich et al. paper (reference 28), while from line 173 to line 177, we used a few sentences to introduce the general approach mostly used in the studies discussed below.

From line 219 to line 216, we reported the study of Cui et al. (reference 22), and from line 219 to on, we discussed the study of Spisni et al. (reference 33).

No other studies are discussed, or not referenced.

Specific concerns

  1. Under Table 1, please detail all Abbrev. Used in the table, according to the Instructions for authors. Same for Table 2.

Thank you for this suggestion. We agree with the reviewer, and we added all acronyms under tables 1 and 2 as suggested.

  1. Please better describe the process of engineered bacteria and the modalities of this process; how it can be technically done?

We thank the reviewer for raising this point as we think adding these details is an important addition to the paper (P 3 line 108). However, advances in the use of probiotics and synthetic biology have led to the development of highly specific genetic engineering methods that differ depending on whether probiotic is used as an intestinal biosensor or live biotherapeutic, inducible or constitutive systems, making their production more efficient and extremely selective. Thus, we added a general description that encompasses all studies listed in the following section in which plasmids are the fundamental tool of recombinant DNA technology engaged in the bioengineering of probiotics. Also, we added the new Figure 1 that shows such general genetic engineering method (P2 line 58).

  1. Please detail the role of certain natural compounds that can be delivered together with the probiotic and decrease the pro-inflammatory markers, such as certain plant-based compounds or supplements that act as prebiotics - in this regard, I suggest checking and referring to https://doi.org/10.3390/diagnostics11061090 https://doi.org/10.3390/microorganisms9030618.

Thank you for bringing this to our attention. We agree with the reviewer that this point should be discussed in our review, and we included comments on the suggested references (P 13 line 455). As we mentioned above, we focused on engineered probiotics only, and no studies tested pre/probiotics formulations that include genetically modified bacteria. Thus, we added a short comment displaying the potential of these formulations in the IBD therapy.

  1. Some figures must be done, to summarize main sections, to increase the impact of your article, and its aspect as well. Detail the mechanisms also.

We thank the reviewer for their thoughts on how to improve the quality of our manuscript. We added an additional figure to summarize the main concepts that have been hit in our review (Figure 1).

  1. A special figure should be made regarding the potential side effect of these modified bacteria in IBD or for the general health.

Thanks for this suggestion. We included the figure as suggested (Figure 3; P14 line 475).

Round 2

Reviewer 2 Report

The authors responded to my requests.